# Conditional Augmentation Enables Effective Use of Synthetic Data in Diffusion Models

## Abstract

Synthetic data generated by foundation models has recently emerged as a promising resource for acquiring pre-trained knowledge and improving data efficiency, especially in scenarios where real data is limited. However, directly incorporating synthetic data often introduces distributional bias and can even result in model collapse. To address these challenges, we propose Conditional Augmentation with Synthetic Data (CASD), a framework guided by three core principles: (a) No harm and no bias: The use of synthetic data should neither degrade model performance nor introduce distributional bias; (b) Positive utility: Synthetic data should enhance model performance; (c) Broad adaptability: The approach should be applicable to synthetic data from diverse sources without requiring case-specific modifications. CASD leverages both real and synthetic data by conditioning on their source labels, treating them as related but distinct domains. This design enables the model to harness large-scale synthetic data to strengthen representation learning, while mitigating bias by focusing on the conditional distribution. During sampling, the model utilizes the enhanced representation function to extract useful information from synthetic data, but fixes the source label to the real domain, ensuring consistency with the target distribution. Experimental results demonstrate that CASD can effectively utilize synthetic data from various foundation models, consistently improving both the quality and diversity of generated images without inheriting distributional bias.

## 1 Introduction

High-quality real-world data has been rapidly consumed by recent foundation models like Llama (Grattafiori et al., 2024) and Stable Diffusion (Esser et al., 2024), raising concerns about the data scarcity crisis that threatens future AI development. In response, synthetic data which refers to data generated algorithmically via simulation or through pretrained generative models (Jordon et al., 2022), has emerged as a promising alternative to support continued model scaling and deployment. Compared to traditional data augmentation methods which add noise to real data or make interpolation between real data (Chawla et al., 2002; Shorten & Khoshgoftaar, 2019), synthetic data generated by the pre-trained foundation model contains additional information from the vast training corpus (Shen et al., 2023; Tian & Shen, 2025). Despite this promise, recent studies (Martínez et al., 2023; Shumailov et al., 2024; Bohacek & Farid, 2023; Guo et al., 2024; Alemohammad et al., 2024; Dohmatob et al., 2025) have highlighted serious risks in training on synthetic data, which could cause distribution shift and model collapse in generative models. These findings lead to an essential question: how can synthetic data be leveraged effectively without compromising model quality and reliability?

Several approaches have sought to answer this question through data augmentation. Martínez et al. (2023) attempted to augment real data with synthetic data to prevent model collapse in recursive training. Following this, Shumailov et al. (2023) and Alemohammad et al. (2024) empirically demonstrated that models exclusively trained on the synthetic data degenerate to fixed points and continuously adding fresh real data could prevent the degeneration. Feng et al. (2025) introduced a verifier to select reliable synthetic data for data augmentation, while Jiang et al. (2025) investigated the use of synthetic data from the perspective of transfer learning. Although these studies demonstrate the potential of synthetic data augmentation, they heavily relied on sufficient fresh real data (Fu et al., 2024; Dohmatob et al., 2025), which may not be feasible in practice. Moreover, sim-

ply augmenting with synthetic data may yield limited gains and inevitably introduce distributional biases into the training process.

In the context of large language models(LLMs), recent studies have focused on leveraging synthetic data generated by LLMs to improve themselves (Tao et al., 2024). Huang et al. (2023) and Zelikman et al. (2024) selected high-quality chain-of-thought (CoT) reasoning processes paired with correct answers from LLMs' outputs and further applied these data to fine-tune them. Wang et al. (2023) used LLMs to generate instructions and filtered them to enhance the training corpus. In these studies, a clear golden standard (e.g., whether an answer is correct) provides a reliable criterion for selecting high-quality synthetic data generated by LLMs. However, for image generation, which is typically evaluated only through perceptual metrics or human preference, such objective standards are much more difficult to establish. How to incorporate synthetic data to enhance generative learning is still a challenge.

To address these limitations, we summarize three basic principles of using synthetic data for generative learning:

(1) **No harm and no bias**: Using synthetic data should not make the model degenerate or introduce distribution bias.

(2) **Positive utility**: Using synthetic data should benefit model performance, such as improved sample quality, diversity, or convergence.

(3) **Broad adaptability**: Methods should apply to synthetic data from diverse sources without requiring case-specific modifications.

Guided by these three core principles, we propose Conditional Augmentation with Synthetic Data (CASD), a general framework for generative learning that leverages conditional diffusion models. CASD treats real and synthetic data as distinct but related sources by introducing source indices as additional conditions, then learning their conditional distribution. This design allows the model to extract informative patterns from large-scale synthetic data and improves the representation learning. During sampling, the source index is fixed to the real domain and this mechanism ensures that the model can maintain distributional consistency and safely facilitate the cross-distributional knowledge transfer without inheriting potential biases from synthetic data.

Our experiments demonstrate that CASD effectively integrates synthetic data from multiple foundation models, yielding substantial gains in both image quality and diversity. Furthermore, when real data are scarce, CASD mitigates duplication and replication of training examples in generative modeling, as documented by prior work (Feng et al., 2021; Somepalli et al., 2023).

In summary, our contributions are

- We articulate three fundamental principles that formalize the systematic use of synthetic data in generative learning.
- We introduce CASD, a general framework that conditions on source identifiers to leverage synthetic data while preserving distributional consistency for real-world tasks.
- We present comprehensive experiments demonstrating that CASD enhances model performance and generalization, particularly in low-real-data regimes.

## 2 BACKGROUND

In this section, we survey related work on synthetic data and briefly overview diffusion models from the perspective of denoising score matching.

### 2.1 SYNTHETIC DATA

Synthetic data from pre-trained models has become a valuable tool for addressing data scarcity and privacy constraints. In generative learning and large language models, self-evolution, which trains models on their own outputs, has recently gained significant traction. While naively using synthetic data can introduce errors and even trigger model collapse (Bohacek & Farid, 2023; Alemohammad et al., 2024; Guo et al., 2024; Shumailov et al., 2024), carefully curated synthetic data can markedly

improve performance, revealing the promise of self-improving large models (Huang et al., 2023; Wang et al., 2023; Zelikman et al., 2024; Feng et al., 2025; Jiang et al., 2025).

A central challenge is the distributional gap between real data and foundation-model outputs. To mitigate these biases, Shen et al. (2023) fine-tuned a pre-trained model and generated task-specific synthetic data, with particular benefits under distributional imbalance (Tian & Shen, 2025). Angelopoulos et al. (2023a;b) used pre-trained models to annotate unlabeled data and incorporated debiasing terms to ensure valid statistical inference. In related directions, McCaw et al. (2024) integrated synthetic data via surrogate variables to improve genome-wide association studies, and Liu et al. (2024) used synthetic data to estimate the distribution of test statistics in hypothesis testing.

Taken together, synthetic data provides a practical mechanism for unlocking and repurposing the knowledge embedded in pre-trained models, with significant potential to address core challenges in machine learning and statistical inference.

## 2.2 DIFFUSION MODEL AND DENOISING SCORE MATCHING

In this paper, we focus on diffusion models for generative learning, a leading class of generative models that achieves state-of-the-art results in high-fidelity image synthesis (Ho et al., 2020; Dhariwal & Nichol, 2021; Rombach et al., 2022). The diffusion process can be formulated via stochastic differential equations (SDEs) (Anderson, 1982; Song et al., 2021):

$$dX_t = f(X_t, t)dt + g(t)dW \tag{1}$$

$$dX_t = [f(X_t, t) - g^2(t)\nabla \log p_t(X_t)]dt + g(t)d\overline{W}, \tag{2}$$

where $W$ is the standard Wiener process, $\overline{W}$ denotes the Wiener process running backward in time, $f(\cdot, t)$ is the drift term, $g(t)$ denotes the diffusion term, $p_t(X_t)$ denotes the density function of $X_t$ and $\nabla \log p_t(X_t)$ is the score function. The forward process Eq. 1 gradually perturbs data $X_0 \sim p_0$ into noise $X_T$ and the reverse process Eq. 2 flows back, transforming the noise $X_T$ back to the target distribution $X_0$.

To learn the reverse process, one learns a neural network $s_\theta(x_t, t)$ to estimate the score function via denoising score matching (Hyvärinen, 2005; Vincent, 2011; Song & Ermon, 2019; Song et al., 2021) by minimizing

$$\mathcal{L}(\theta) = \int \mathbb{E}_{x_0 \sim p_0} \mathbb{E}_{x_t \sim p(x_t)} [\lambda(t) ||s_\theta(x_t, t) - \nabla \log p_t(x_t|x_0)||_2^2] dt, \tag{3}$$

where $\lambda(t)$ is a temporal weighting function. Because the real data distribution $P_r$ and the synthetic data distribution $P_s$ can share structural similarities, score-based methods offer a natural mechanism to combine both sources: by consistently estimating score functions, diffusion models can leverage information from synthetic data while mitigating distributional bias.

## 3 METHODOLOGY

In this section, we present the details of our proposed CASD method. Section 3.1 introduces preliminaries, and Section 3.2 describes the CASD method.

## 3.1 PRELIMINARIES

Let $P_r$ denote the distribution of real-world data that we are interested in and $P_s$ be the distribution of synthetic data generated by pretrained foundation models. The training dataset consists of two parts $D_{train} := D_r \cup D_s$:

- Real data $D_r := \{x_i^r\}_{i=1}^n$ are i.i.d. sampled from the real distribution $P_r$, which are high-quality but limited and expensive to obtain. At diffusion time step $t_i$, the $i$-th sample is denoted $x_{i,t_i}^r$.

- Synthetic data $D_s := \{x_j^s\}_{j=1}^m$ are i.i.d. sampled from $P_s$, where $m$ could be much larger than $n$ since synthetic data can be infinitely sampled from the foundation models. At diffusion time step $t_j$, the $j$-th sample is denoted $x_{j,t_j}^s$.

A useful tool in denoising score matching is Tweedie's formula (Robbins, 1992; Efron, 2011), which links score functions with conditional expectations:

**Lemma 3.1 (Tweedie's formula)** *Suppose $X \sim u$ and $\epsilon \sim N(0, I)$. Let $Y = X + \epsilon$ and $p(y)$ be the marginal density of $Y$. Then $\mathbb{E}[X|Y = y] = y + \nabla_y \log p(y)$.*

This identity provides a way to empirically estimate the score function and serves as the basis of our approach.

## 3.2 Conditional Synthetic Data Augmentation

The key intuition of CASD is that real and synthetic data share structural similarities but remain different; the model should both exploit their commonalities and learn to distinguish their sources. Accordingly, CASD tags each training example $x$ with a source label $y$ (e.g., real vs. synthetic model index), yielding an augmented training dataset

$$\widetilde{D}_{train} := \{(x_i^r, y_r)\}_{i=1}^n \cup \{(x_j^s, y_s)\}_{j=1}^m,$$

where $y_r = 0$ for real data and $y_s = 1$ for synthetic data. The corresponding joint distribution $\widetilde{P}_{XY}$ of the training dataset $\widetilde{D}_{train}$ integrates information from real data, synthetic data and the source indicator. The conditional distribution $\widetilde{P}_{X|Y}$ enjoys property $\widetilde{P}_{X|Y=0} = P_r$, ensuring that sampling with $y = 0$ reproduces the real distribution. We then train a conditional diffusion model to learn the conditional distribution $\widetilde{P}_{X|Y=0}$. As shown in Figure 1, the score neural network $s_\theta(x_t, y, t) = f_{\theta_3}(R_{\theta_1}(x_t), e_{\theta_2}(y))$ is parameterized in three components :

- $R_{\theta_1}$: a shared representation network that encodes both real and synthetic samples. Large-scale synthetic data strengthens its representation learning and improves convergence.
- $e_{\theta_2}$: a linear embedding function that maps the one-hot encoded source label $y$ into a high-dimensional embedding space, enabling the model to differentiate between data sources.
- $f_{\theta_3}$: a U-Net (Ronneberger et al., 2015) that combines the data representation $R_{\theta_1}(x)$ and the label embedding $e_{\theta_2}(y)$ to approximate the score function.

Formally, $s_\theta$ can be learned by minimizing the following objective function:

$$\widetilde{\mathcal{L}}(\theta) = \int \mathbb{E}_{(x_0,y) \sim \widetilde{P}_{XY}} \mathbb{E}_{x_t \sim p(x_t|x_0)} [\lambda(t) ||s_\theta(x_t, y, t) - \nabla_{x_t} \log p_t(x_t|x_0)||_2^2] dt. \quad (4)$$

According to Tweedie's formula in Lemma 3.1, the score function $\nabla_{x_t} \log p_t(x_t|x_0)$ can be approximated with $x_0 - x_t$, and the empirical loss function is

$$\widetilde{\mathcal{L}}_n(\theta) = \frac{1}{n+m} \sum_{i=1}^n \lambda(t_i) ||f_{\theta_3}(R_{\theta_1}(x_{i,t_i}^r), e_{\theta_2}(0)) - (x_i^r - x_{i,t_i}^r)||_2^2$$

$$+ \frac{1}{n+m} \sum_{j=1}^m \lambda(t_j) ||f_{\theta_3}(R_{\theta_1}(x_{j,t_j}^s), e_{\theta_2}(1)) - (x_j^s - x_{j,t_j}^s)||_2^2. \quad (5)$$

At the sampling stage, the reverse diffusion process is discretized and solved using the 2nd order Heun method (Heun et al., 1900; Ascher & Petzold, 1998; Karras et al., 2022). At each step, Heun's method first computes a provisional update at the current point using the simple Euler method, and then refines the update by averaging it with a correction term which is calculated after the provisional update. This yields a second-order accurate approximation that improves stability and accuracy compared to the Euler method. In this procedure, the score function is replaced by $s_\theta(X_t, 0, t)$, and the source label $Y$ is fixed to 0, which guarantees that generations follow the real distribution while still benefiting from synthetic-informed representations. Details of the CASD algorithm are presented in Algorithm 1.

## 4 Experiments

In this section, we evaluate the proposed method. Section 4.1 describes the experimental setup, and the subsequent subsections present the results.

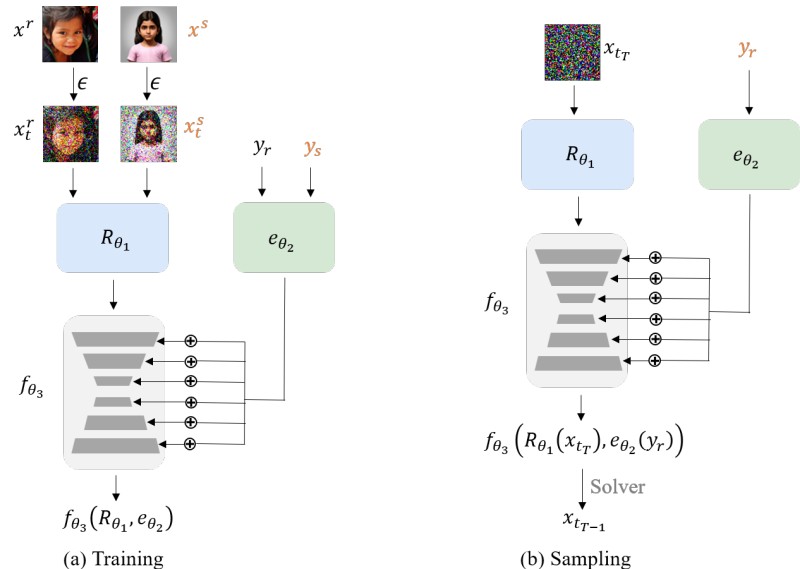

Figure 1: Overview of CASD framework during training (a) and sampling (b).

---

**Algorithm 1 Conditional Augmentation with Synthetic Data**

---

1: **Input:** Real data $D_r = \{x_i^r\}_{i=1}^n$, synthetic data $D_s = \{x_j^s\}_{j=1}^m$
2: **Training stage:**
3: Assign source labels: $\widetilde{D}_r = \{(x_i^r, y_r)\}_{i=1}^n$, $\widetilde{D}_s = \{(x_j^s, y_s)\}_{j=1}^m$
4: Initialize score network $s_\theta$ with parameters $\theta$
5: Train $s_\theta$ by minimizing the weighted conditional score-matching loss in Eq. 5
6: Obtain trained model $s_{\theta^*}$
7: **Sampling stage:**
8: Fix source label $Y = y_r = 0$ (real domain)
9: Starting from noise $x_{t_T} \sim \mathcal{N}(0, I)$, simulate the reverse SDE

$$dX_t = \left[f(X_t, t) - g^2(t)\, s_{\theta^*}(X_t, 0, t)\right]dt + g(t)\, d\overline{W}$$

for $T$ steps using the $2^{\text{nd}}$ order Heun method
10: **Output:** Generated samples $\{\widetilde{x}_i\}_{i=1}^N$

---

## 4.1 EXPERIMENTAL SETUP

**Datasets.** We evaluate CASD on two standard generative modeling benchmarks: CIFAR-10 (Krizhevsky et al., 2009) at its native $32 \times 32$ resolution and FFHQ (Karras et al., 2019), down-sampled from $1024 \times 1024$ to $64 \times 64$. For the real dataset $D_r$, we randomly sample n examples from the full training set. For the synthetic dataset $D_s$, we generate m samples from pretrained foundation models using distinct random seeds.

**Models.** We follow the EDM implementation (Karras et al., 2022) in training the conditional diffusion model of CASD, which is a widely adopted framework for high-quality image synthesis. All architectural and training hyperparameters are kept consistent across methods and detailed parameters are presented in Appendix A.2.

**Evaluation metrics.** We evaluate the results from three aspects: Fréchet Inception Distance (Heusel et al., 2017, FID) is used to measure the quality of generative images; recall (Sajjadi et al., 2018; Kynkäänniemi et al., 2019) and convergence (Naeem et al., 2020) are applied to evaluate the diversity; SSIM(Wang et al., 2004) and LPIPS (Zhang et al., 2018) are adopted to measure memorization and replication. All these metrics are calculated on 50,000 generated samples.

**Compared methods.** We compare CASD against two methods:

- Diffusion Model (DM): An unconditional diffusion model trained only on real data $D_r$;

- Synthetic Data Augmentation (SDA): An unconditional diffusion model trained jointly on real and synthetic data $D_r \cup D_s$.

All methods use the same architecture and follow the EDM training protocol.

## 4.2 Expanding Effective Data with Synthetic Samples

We first consider the setting where synthetic models are pre-trained on datasets following the real distribution $P_r$, so that the synthetic distribution $P_s$ is close to $P_r$. For CIFAR-10 and FFHQ, we construct synthetic datasets using StyleGAN (Karras et al., 2019; 2020) and EDM (Karras et al., 2022).

We fix the synthetic dataset size to 10,000 (abbreviated as 10k) and vary the number of real samples $n$ from 1,000 to 10,000 (abbreviated as 1k–10k). As shown in Table 1, CASD almost achieves lower FIDs than both DM and SDA across datasets and synthetic models. Notably, with synthetic data from EDM, CASD trained on only 2,000 real CIFAR-10 samples matches the result of the DM method trained on 10,000 real samples. Similarly, on FFHQ, CASD with 1,000 real samples outperforms the DM method trained on 10k samples. These results show that CASD efficiently leverages information in synthetic data and greatly increases the amount of effective data, effectively improving the model performance. In contrast, SDA is sensitive to the quality of synthetic data and even performs worse than DM when incorporating synthetic data generated by the StyleGAN model. As shown by the gray cells in Table 1, although DM trained on only 1,000 real FFHQ samples exhibits a lower FID than DM trained on 2,000 samples, it actually suffers from model collapse and only memorizes these training samples (Feng et al., 2021). This phenomenon will be discussed in Section 4.4.

Next, we fix the number of real samples $n$ to 2,000 and vary the synthetic samples $m$ from 0 to 10,000 in the FFHQ dataset. As shown in Figure 2, CASD steadily improves the FID as $m$ increases and consistently surpasses SDA using both synthetic datasets. These results demonstrate that leveraging synthetic data from models pre-trained on the same dataset can enhance generative learning and achieve better results with limited real-world data. Importantly, unlike SDA, CASD successfully avoids inheriting the distributional bias when synthetic data is generated from StyleGAN. Moreover, CASD effectively identifies useful patterns from synthetic data and integrates them with real data, thereby expanding the model's generalization capability and improving overall performance.

| Synthetic Model | | StyleGAN | | | | EDM | | | |
|---|---|---|---|---|---|---|---|---|---|
| Real Dataset | $n$ | 1k | 2k | 5k | 10k | 1k | 2k | 5k | 10k |
| | DM | 10.68 | 9.62 | 5.69 | **3.11** | 10.68 | 9.62 | 5.69 | 3.11 |
| CIFAR-10 | SDA | **5.38** | 4.98 | 4.20 | 3.51 | 3.86 | 3.70 | 3.24 | 2.90 |
| | CASD | 5.47 | **4.66** | **3.83** | 3.27 | **3.61** | **3.11** | **2.65** | **2.37** |
| | DM | 8.83 | 12.70 | 6.60 | 4.48 | 8.83 | 12.70 | 6.60 | 4.48 |
| FFHQ | SDA | 26.03 | 23.53 | 16.38 | 11.36 | 5.53 | 5.21 | 4.77 | 4.17 |
| | CASD | **6.26** | **5.93** | **5.09** | **4.14** | **4.26** | **4.25** | **3.66** | **3.31** |

Table 1: FID ($\downarrow$) with varying sizes of real data $n$ when the synthetic dataset size is fixed to 10k. CASD consistently improves over the DM and SDA, especially under limited real data. Cells shaded in gray indicate that model collapse happens.

## 4.3 Robustness to Synthetic Distribution Mismatch

In this subsection, we analyze the impact of the distribution of synthetic data $P_s$ when it does not perfectly align with the real distribution $P_r$. Specifically, we use a subset of CIFAR-10 as the real dataset, consisting of the first 8 classes. We then construct synthetic datasets under three different cases, where the synthetic model is pre-trained on: (1) all 10 classes, (2) the last 8 classes, and (3) the middle 6 classes of CIFAR-10. As a result, the support of $P_s$ either contains, is contained within, or intersects with the support of $P_r$. For each case, we use 1,000 real samples that consist of 125 samples per class and 10,000 synthetic samples to train the generative models. Table 2 shows that CASD consistently and significantly improves model performance across all three cases, achieving

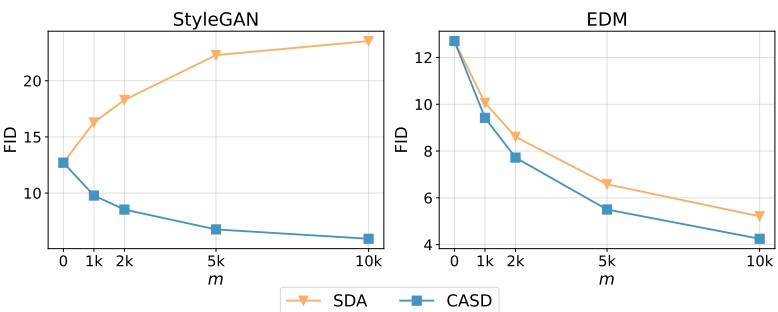

Figure 2: FID ($\downarrow$) as the size of synthetic data $m$ varies from 0 to 10,000 with the real dataset size fixed at $n = 2,000$ on FFHQ dataset. CASD consistently outperforms SDA across both synthetic data sources and maintains performance even with lower-quality synthetic data.

much lower FIDs compared to both DM and SDA. In particular, SDA often fails when $P_s$ deviates from $P_r$, whereas CASD remains robust.

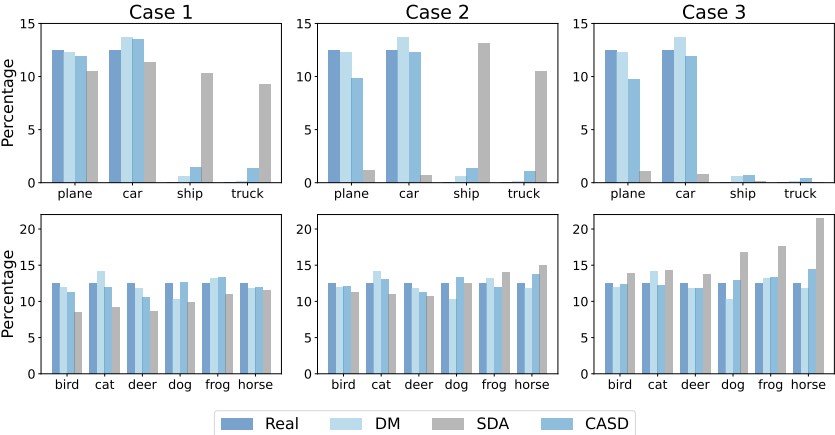

Figure 3: Approximate class distributions of generated images in the three mismatch cases, obtained by a classifier trained on CIFAR-10 dataset. Top row: four special categories (plane, car, ship, truck) that differ between $P_r$ and $P_s$. Bottom row: six common categories shared across $P_r$ and $P_s$. CASD aligns closely with $P_r$ while SDA are heavily influenced by $P_s$.

To gain a deeper understanding of these results, we approximate the distribution of generated images using a classifier that is trained on the CIFAR-10 dataset. As shown in Figure 3, the first two categories in the first row (plane and car) are included in $D_r$ but excluded from some synthetic datasets, while the last two categories (ship and truck) are excluded from $D_r$ but present in synthetic datasets. The second row shows the proportions of the middle six categories, which are present in both $D_r$ and all synthetic datasets.

Compared to DM, the distribution of SDA is strongly distorted by $P_s$ and shows significant differences with the real data, highlighting its sensitivity to distribution mismatch between $P_r$ and $P_s$. In contrast, CASD precisely identifies the overlapped area in support and leverages useful information from synthetic data, suppressing bias from irrelevant categories. Furthermore, the conditional sampling mechanism ensures that generated images remain consistent with $P_r$.

| Methods | Case 1 | Case 2 | Case 3 |
|---------|--------|--------|--------|
| DM      | 11.28  | 11.28  | 11.28  |
| SDA     | 9.98   | 12.77  | 13.71  |
| CASD    | **4.10** | **4.59** | **4.79** |

Table 2: The table shows FIDs($\downarrow$) of three methods in different cases, where the support of $P_s$ either contains, is contained within, or intersects with the support of the real data distribution $P_r$.

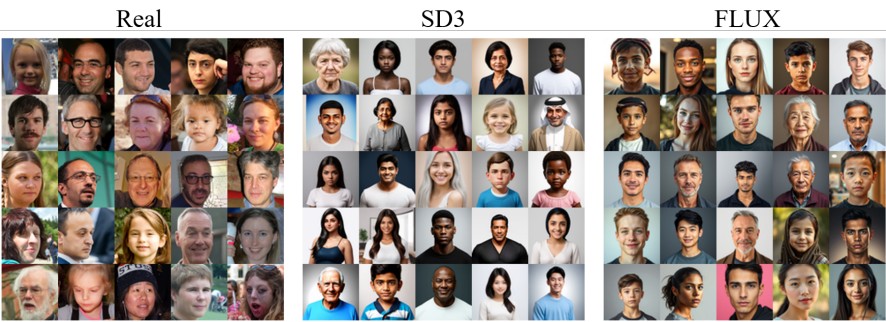

Figure 4: From left to right, the figure shows images sampled from the real FFHQ dataset, images generated by SD3, and images generated by FLUX

## 4.4 MITIGATING REPLICATION AND IMPROVING DIVERSITY

In this subsection, we study a realistic setting where synthetic data are produced by popular foundation models—Stable Diffusion 3 (SD3) (Esser et al., 2024) and FLUX (Labs et al., 2025). Both are pretrained on large, diverse corpora and generate images from text prompts. As illustrated in Figure 4, the outputs of SD3 and FLUX exhibit visual styles that differ noticeably from real FFHQ images, creating a substantial distribution shift and making it challenging to integrate these synthetic samples effectively.

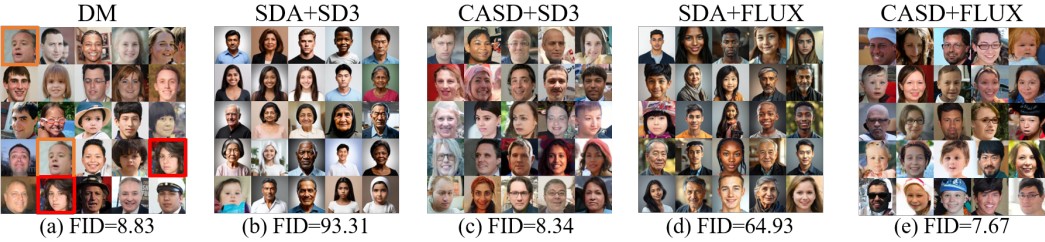

Figure 5: DM exhibits limited diversity with noticeable duplication (highlighted by red boxes and orange boxes in (a)) while the SDA model generates images with styles similar to the synthetic data (shown in (b) and (d)). In contrast, the CASD model effectively enhances diversity while preserving style consistency (shown in (c) and (e)).

Figure 5 shows images generated by models that use 1,000 real samples and 10,000 synthetic samples. SDA fails to handle the style differences between real data and synthetic data, generating images that resemble the synthetic data and leading to higher FIDs than other methods. In contrast, CASD not only successfully maintains the distribution consistency and generates images with the target FFHQ style, but it also leverages useful information from the synthetic data and slightly improves the quality of generated images using both synthetic datasets.

DM       CASD+SD3       CASD+FLUX

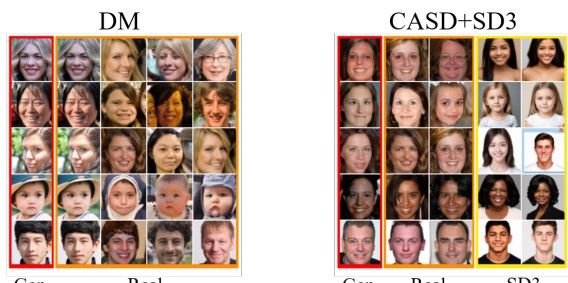 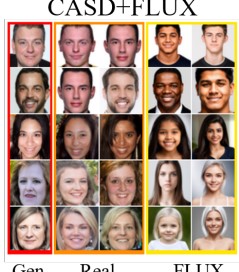

Gen    Real       Gen    Real    SD3      Gen    Real    FLUX

Figure 6: DM exhibits pronounced replication of real data, while incorporating synthetic data by CASD from SD3 or FLUX effectively suppresses these phenomena.

Another limitation of DM is duplication and replication. While its outputs appear sharp and realistic, the model frequently synthesizes near-identical images, highlighted by the red and orange boxes in Figure 5. Furthermore, Figure 6 shows almost exact copies of training samples among DM's generations. These findings suggest that, with limited real data, the model overfits, memorizing the training set and reproducing it rather than truly generating new samples from the underlying data distribution. This behavior is well documented when training data are scarce (Feng et al., 2021; Somepalli et al., 2023). Incorporating synthetic data via CASD mitigates this issue by reducing memorization and promoting greater sample diversity.

| Methods | Recall↑ | Coverage↑ | LPIPS↑ | SSIM↓ |
|---------|---------|-----------|--------|-------|
| DM | 0.3768 | 0.8037 | 0.0445 | 0.8429 |
| CASD+SD3 | 0.5112 | 0.8691 | 0.2261 | 0.2443 |
| CASD+FLUX | 0.5351 | 0.8806 | 0.2265 | 0.2426 |

Table 3: CASD significantly improves Recall, Coverage, and LPIPS while reducing SSIM, demonstrating that it improves diversity and mitigates replication.

As shown in Table 3, CASD substantially improves the diversity of generated images. Higher Recall and Coverage indicate that CASD captures a broader portion of the real data distribution. Moreover, increased LPIPS and reduced SSIM suggest that the generator explores the distribution more widely and generalizes to regions absent from the limited training set. These results underscore the value of integrating synthetic data via CASD: it not only mitigates duplication and replication, but also helps models trained on scarce real data more faithfully approximate the target distribution.

## 5   CONCLUSION AND DISCUSSION

We propose three guiding principles for using synthetic data in generative learning and introduce CASD, a framework that leverages synthetic data without degrading model quality. CASD treats real and synthetic data as distinct yet related domains by conditioning on explicit source labels, preventing distributional bias and preserving the target distribution. By jointly learning from both domains, it extracts rich signals from large-scale synthetic data to strengthen representation learning and boost model capacity when real data are scarce. At sampling time, we fix the source label to the real domain to avoid inheriting synthetic-domain bias, while shared representations enable effective knowledge transfer from synthetic to real. Extensive experiments on CIFAR-10 and FFHQ show that CASD consistently improves both image quality and diversity across settings, particularly with limited real data. Following our principles, CASD harnesses the advantages of large-scale synthetic data while avoiding the distribution shift and model collapse that hamper prior approaches.

While our results highlight the promise of CASD, several avenues for future research remain open. First, integrating CASD into self-evolving or self-improving frameworks could shed light on the effects of synthetic data in foundation model training. Second, CASD could be extended to domains where the labeled data is limited or unbalanced, such as semi-supervised learning or medical problems. Finally, developing a deeper theoretical understanding of CASD could yield principled insights into how synthetic and real data interact, which may guide the design of the new synthetic data framework.

ETHICS STATEMENT

This work investigates the use of synthetic data generated by large-scale generative models for improving data efficiency in machine learning. All datasets used in this study are publicly available benchmark datasets (e.g., CIFAR-10, FFHQ) and the synthetic datasets are produced using publicly available pre-trained generative models, with no private data or personally identifiable information involved. Since the synthetic data are derived from well-documented public sources and our method does not create realistic personal information, we believe there are no additional privacy or security concerns beyond standard machine learning practices.

REPRODUCIBILITY STATEMENT.

We have taken extensive steps to ensure that all results in this paper are reproducible. A complete description of the datasets, data preprocessing, and synthetic data generation procedures is provided in Appendix A.1, including examples and FID evaluations of the synthetic data. Model architectures, hyperparameters, and training schedules are detailed in Appendix A.2 and Table 9, including batch size, learning rate, etc. Our sampling algorithm is specified in Algorithm 2 in Appendix A.3.

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

CONTENTS

# A    IMPLEMENTATION DETAILS

## A.1    DATASETS

### A.1.1    REAL DATA

In this paper, we consider the CIFAR-10 dataset (Krizhevsky et al., 2009) and the FFHQ dataset (Karras et al., 2019) as the real data. To mitigate the influence of randomness, we use the first $n$ samples as the real training data $D_r$. Details of these two datasets are presented in Table 4.

|  | CIFAR-10 | FFHQ |
|---|---|---|
| Resolution | $3\times32\times32$ | $3\times 64\times64$ |
| Number of instances | 50,000 | 70,000 |

Table 4: Configurations of CIFAR-10 and FFHQ

### A.1.2    SYNTHETIC DATA

In this paper, we use synthetic data generated by different pre-trained models.

As for experiments conducted in Section 4.2, the synthetic data is generated by the pre-trained Style-GAN models (Karras et al., 2019; 2020) and pre-trained EDM models (Karras et al., 2022) using different random seeds. Specifically, as the officially released StyleGAN model for the CIFAR-10 dataset is a conditional generative model, we sample 5,000 images per class to construct a balanced and comprehensive synthetic dataset. Besides, since the officially released StyleGAN model for the FFHQ dataset is trained on images with $3\times1024\times1024$ resolution, we use it to generate synthetic samples and resize them to $3\times64\times64$ resolution. Moreover, we use the officially released EDM models to unconditionally generate synthetic data for CIFAR-10 and FFHQ with proper resolutions. FIDs of synthetic data generated by pre-trained StyleGAN models and EDM models are shown in Table 5, where the abnormal FID of StyleGAN may be influenced by the downsampling process from 1024×1024 to 64×64 resolution. As presented in Figure 7 and Figure 8, the synthetic data exhibits high quality and closely aligns with the real dataset.

| Synthetic Model | StyleGAN | EDM |
|---|---|---|
| CIFAR-10 | 2.46 | 1.96 |
| FFHQ | 17.49 | 2.37 |

Table 5: FIDs of synthetic data generated by pre-trained StyleGAN models and EDM models.

As for experiments conducted in Section 4.3, since pre-trained models are not available for Cases 2 and Case 3, we use the last 8 classes and the middle 6 classes of CIFAR-10 dataset to train unconditional diffusion models based on the EDM architecture (Karras et al., 2022), respectively. Then these two models are employed to generate synthetic data and FIDs are presented in Table 6. Examples of synthetic data for three cases are illustrated in Figure 9.

|  | Case 1 | Case 2 | Case 3 |
|---|---|---|---|
| FID | 1.96 | 2.44 | 2.89 |

Table 6: FIDs of synthetic data for three Cases.

As for experiments conducted in Section 4.4, the synthetic data is generated by Stable Diffusion 3 (Esser et al., 2024, SD3) and FLUX (Labs et al., 2025) through text-to-image generation. To construct comprehensive synthetic datasets, we employ LLMs to generate various prompts, which are then used to generate images. Some examples of prompts are provided in Table 7 and some examples of the generated images are shown in Figure 10 and Figure 11. These synthetic images

Real StyleGAN EDM

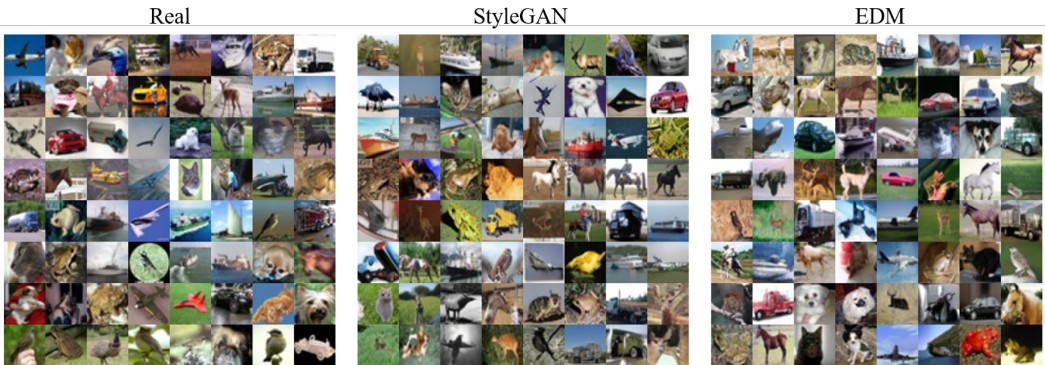

Figure 7: From left to right, the figure shows images sampled from the real CIFAR-10 dataset, synthetic CIFAR-10 images generated by StyleGAN, and synthetic CIFAR-10 images generated by EDM.

Real StyleGAN EDM

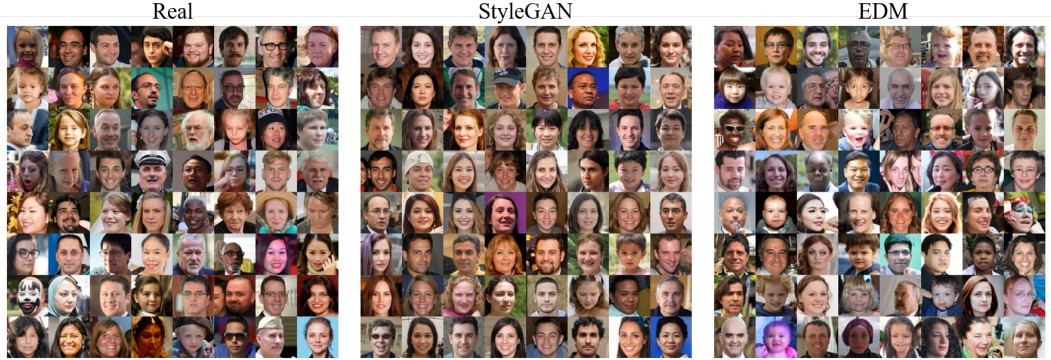

Figure 8: From left to right, the figure shows images sampled from the real FFHQ dataset, synthetic FFHQ images generated by StyleGAN, and synthetic FFHQ images generated by EDM.

Case 1 Case 2 Case 3

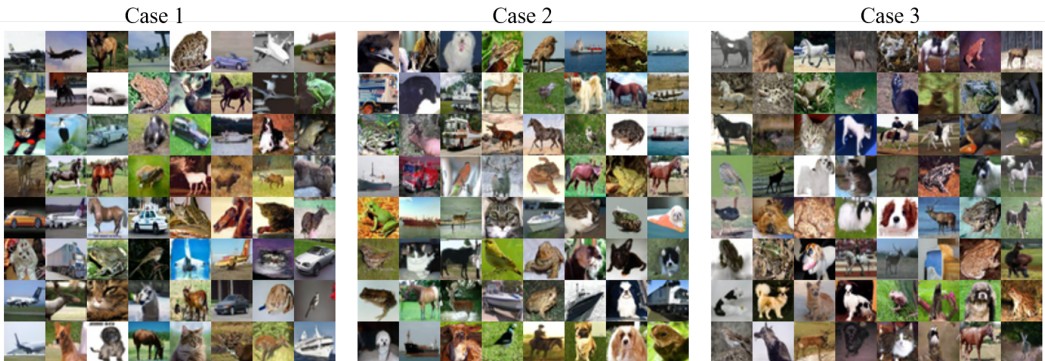

Figure 9: From left to right, the figure shows synthetic images for Case 1, Case 2 and Case 3.

| A DSLR photo of a child Asian woman with a confident look, studio background, headshot, photorealistic, high detail |
| --- |
| A DSLR photo of a teen Black man with a gentle smile, plain background, centered composition, photorealistic, high detail |
| A DSLR photo of a middle-aged Latino woman with a confident look, studio background, realistic lighting, photorealistic, high detail |
| A DSLR photo of an elderly Middle Eastern man with a gentle smile, plain background, close-up face, photorealistic, high detail |
| A passport-style photo of a young South Asian woman with a neutral expression, subtle gradient background, close-up face, photorealistic, high detail |
| A passport-style photo of an elderly White man with a neutral expression, studio background, sharp focus, photorealistic, high detail |
| A passport-style photo of a middle-aged Black woman with a gentle smile, plain background, centered composition, photorealistic, high detail |
| A professional headshot of a teen Asian woman with a serious face, studio background, close-up face, photorealistic, high detail |
| A professional headshot of a young Latino man with a confident look, studio background, centered composition, photorealistic, high detail |
| A professional headshot of an elderly South Asian woman with a gentle smile, subtle gradient background, headshot, photorealistic, high detail |
| A realistic portrait of a child Middle Eastern woman with a confident look, studio background, close-up face, photorealistic, high detail |
| A realistic portrait of a middle-aged White man with a neutral expression, subtle gradient background, headshot, photorealistic, high detail |
| A realistic portrait of a young Black woman with a confident look, plain background, realistic lighting, photorealistic, high detail |
| A studio portrait of a teen South Asian woman with a serious face, indoor background, realistic lighting, photorealistic, high detail |
| A studio portrait of an elderly Latino woman with a gentle smile, plain background, sharp focus, photorealistic, high detail |
| A studio portrait of a child White man with a serious face, plain background, realistic lighting, photorealistic, high detail |

Table 7: Examples of prompts.

are resized to $3\times64\times64$ resolution and their FIDs are shown in Table 8. Although there are significant differences between real data and synthetic data, the synthetic data still retains some useful information, such as facial structure.

|  | SD3 | FLUX |
| --- | --- | --- |
| FID | 109.97 | 71.10 |

Table 8: FIDs of synthetic data generated by SD3 and FLUX.

## A.2 MODEL CONFIGURATION

Our model employs a conditional denoising architecture based on the EDM (Karras et al., 2022) framework and we decompose the architecture into three main components: the representation functions for input images, the embedding functions for conditioning signals, and a U-Net backbone network.

**Representation Function** $R_{\theta_1}$. We use a 3×3 convolutional layer with padding of 1 to map the images with 3 RGB channels to 128 base feature channels.

**Embedding Function** $e_{\theta_2}$. The source label $Y$ is first embedded into a 128-dimensional vector and then processed through a two-layer MLP network with SiLU activation, resulting in a 512-dimensional representation.

**U-Net** $f_{\theta_3}$. We use the U-Net (Ronneberger et al., 2015) architecture to estimate both the score function, which consists of residual blocks, upsampling blocks, downsampling blocks and cross-

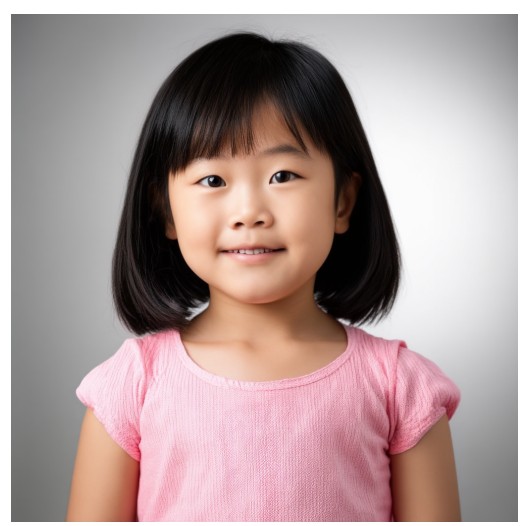

(a) A DSLR photo of a child Asian woman with a confident look, studio background, headshot, photorealistic, high detail

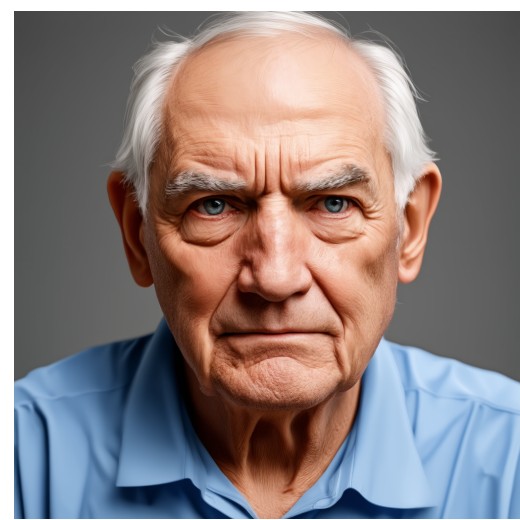

(b) A passport-style photo of an elderly White man with a neutral expression, studio background, sharp focus, photorealistic, high detail

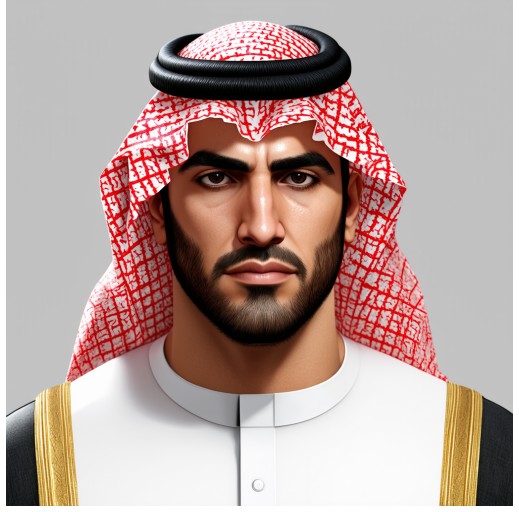

(c) A realistic portrait of a young Middle Eastern man with a serious face, studio background, headshot, photorealistic, high detail

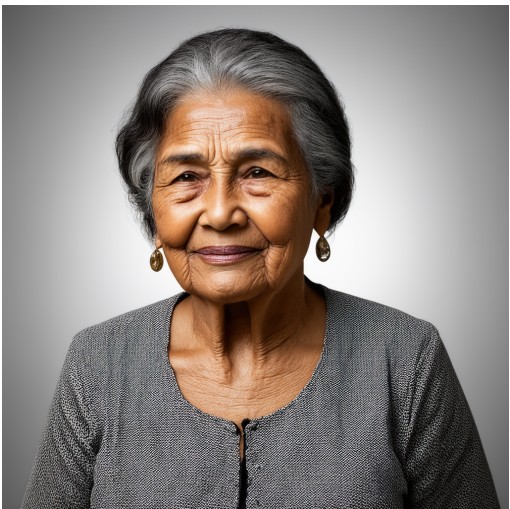

(d) A studio portrait of an elderly Latino woman with a gentle smile, subtle gradient background, headshot, photorealistic, high detail

Figure 10: Examples of images generated by SD3.

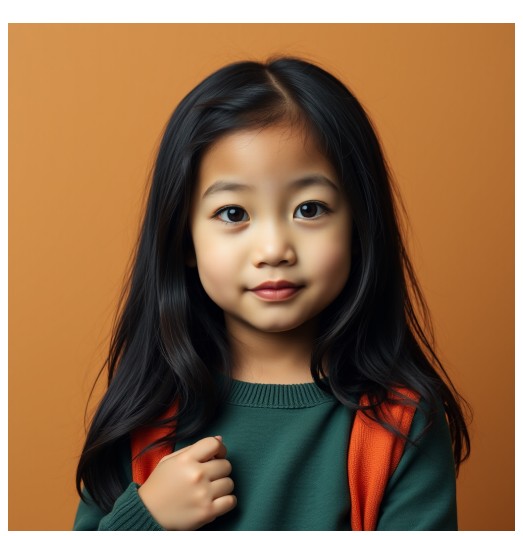

(a) A DSLR photo of a child Asian woman with a confident look, studio background, headshot, photorealistic, high detail

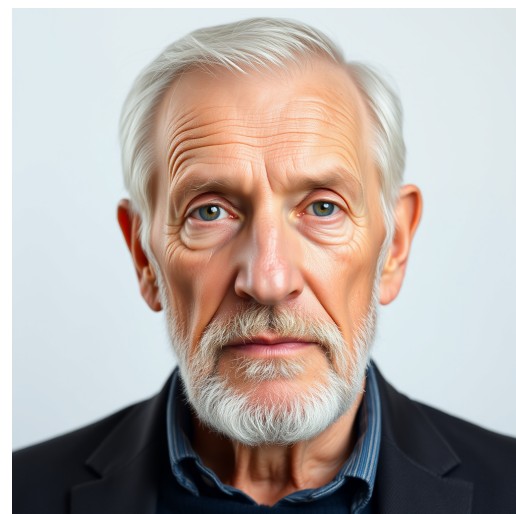

(b) A passport-style photo of an elderly White man with a neutral expression, studio background, sharp focus, photorealistic, high detail

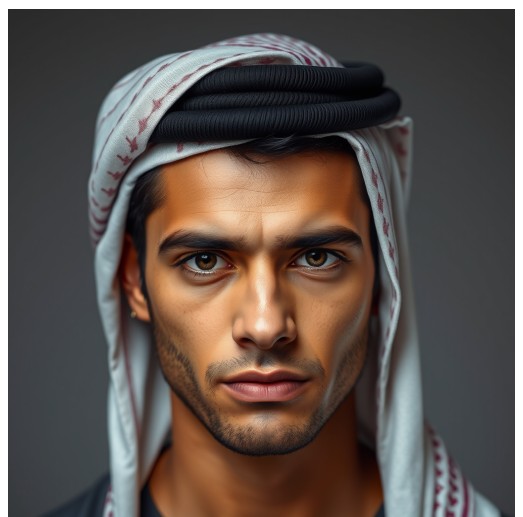

(c) A realistic portrait of a young Middle Eastern man with a serious face, studio background, headshot, photorealistic, high detail

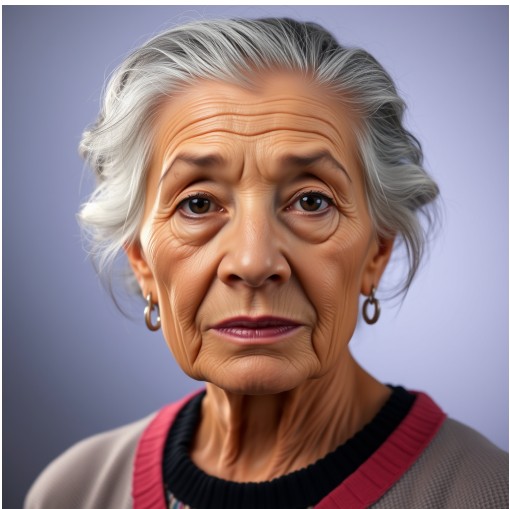

(d) A studio portrait of an elderly Latino woman with a gentle smile, subtle gradient background, headshot, photorealistic, high detail

Figure 11: Examples of images generated by FLUX.

|  | CIFAR-10 | FFHQ |
|---|---|---|
| **Network Architecture** | | |
| U-Net | DDPM++ | DDMP++ |
| Channels per resolution | [2,2,2] | [1,2,2,2] |
| **Training Parameters** | | |
| Learning rate | 0.001 | 0.0002 |
| Total kimg | 60 | 60 |
| Batch size | 512 | 512 |
| Mixed-precision | True | True |
| Dropout probability | 0.13 | 0.05 |
| Augment probability | 0.12 | 0.15 |
| **Sampling** | | |
| Solver | $2^{\text{nd}}$ order Heun | $2^{\text{nd}}$ order Heun |
| NFE | 35 | 79 |

Table 9: Model Configuration

attention layers. For CIFAR-10, the model encompasses 6 feature map resolutions ranging from 32×32 to 8×8 with network widths in sequence (128, 256, 256, 256). For FFHQ, the model encompasses 7 feature map resolutions ranging from 64×64 to 8×8 with network widths in sequence (128, 128, 256, 256, 256). Details of model architectures and parameters are given in Table 9.

### A.3 Sampling Algorithm

---

**Algorithm 2 ODE sampling of $2^{\text{nd}}$ order Heun method**

---

1: **Input:** score function $s_{\theta^*}$, time steps $\{t_i\}_{i=0}^T$ with $t_T = T > t_{T-1} > ... > t_0 = 0$
2: Fix source label $Y$ to 0 and starting from noise $x_{t_T} \sim \mathcal{N}(0, I)$
3: **for** $i \in \{T, T-1 \ldots, 1\}$ **do**
4:     $d_i \leftarrow (x_{t_i} - s_{\theta^*}(x_{t_i}, 0, t_i))/t_i$
5:     $x_{t_{i-1}} \leftarrow x_{t_i} + (t_{i-1} - t_i)d_i$
6:     **if** $t_{i-1} \neq 0$ **then**
7:         $d_i' \leftarrow (x_{t_{i-1}} - s_{\theta^*}(x_{t_{i-1}}, 0, t_{i-1}))/t_{i-1}$
8:         $x_{t_{i-1}} \leftarrow x_{t_i} + (t_{i-1} - t_i)\left(\frac{1}{2}d_i + \frac{1}{2}d_i'\right)$
9:     **end if**
10: **end for**
11: **Return: generated images** $\{\widetilde{X}_i\}_{i=1}^N$

---

## B  Sample Comparison

DM SDA CASD

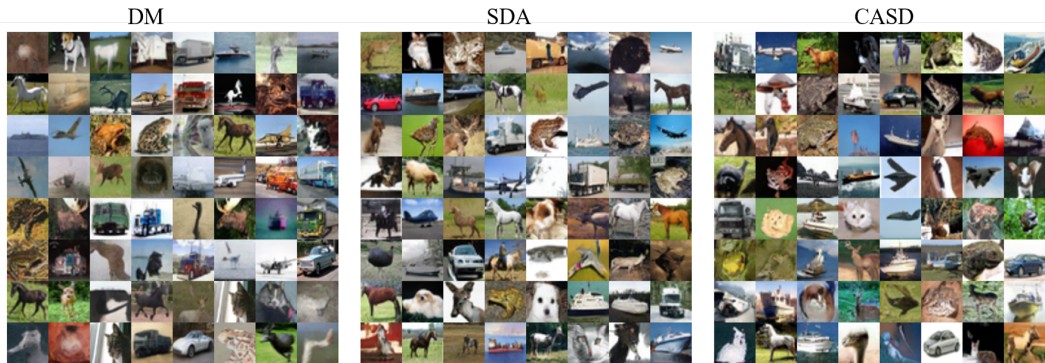

Figure 12: Generated samples from DM, SDA, and CASD trained on 1,000 real CIFAR-10 images and 10,000 synthetic images generated by the pre-trained StyleGAN model.

DM SDA CASD

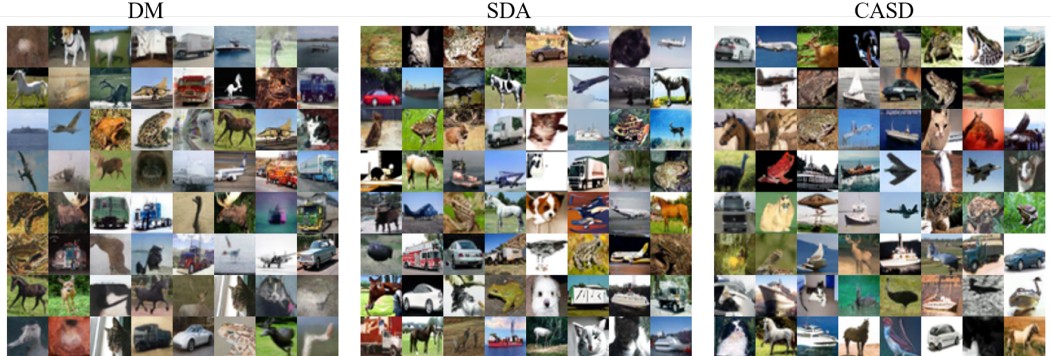

Figure 13: Generated samples from DM, SDA, and CASD trained on 1,000 real CIFAR-10 images and 10,000 synthetic images generated by the pre-trained EDM model.

DM SDA CASD

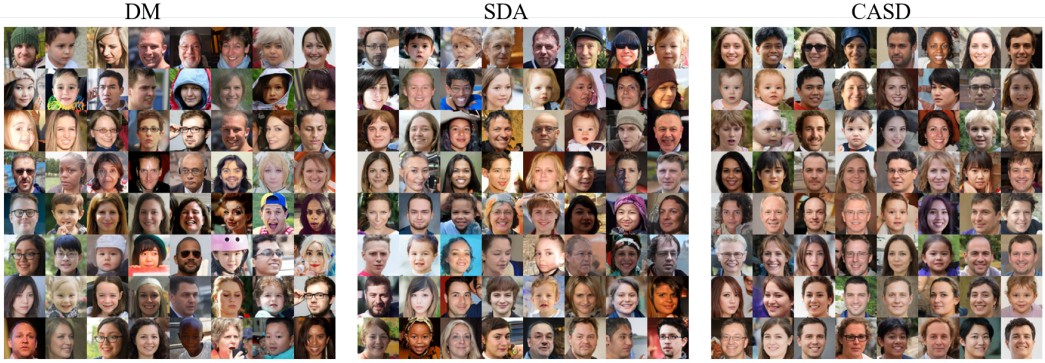

Figure 14: Generated samples from DM, SDA, and CASD trained on 1,000 real FFHQ images and 10,000 synthetic images generated by the pre-trained StyleGAN model.

DM SDA CASD

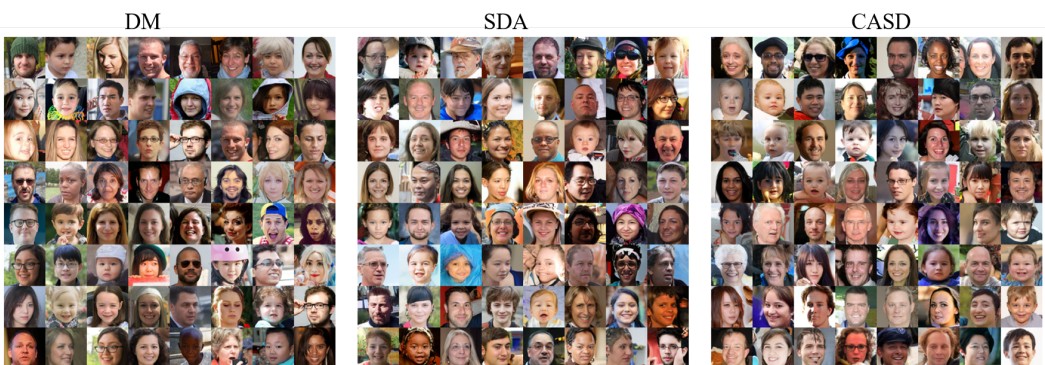

Figure 15: Generated samples from DM, SDA, and CASD trained on 1,000 real FFHQ images and 10,000 synthetic images generated by the pre-trained EDM model.

DM SDA CASD

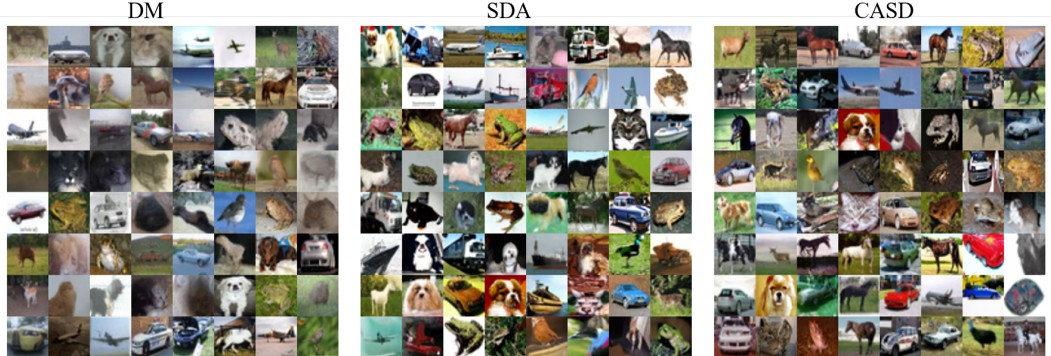

Figure 16: Generated samples from DM, SDA, and CASD trained with 1,000 real images and 10,000 synthetic images (Case 1 in Section 4.3)

DM SDA CASD

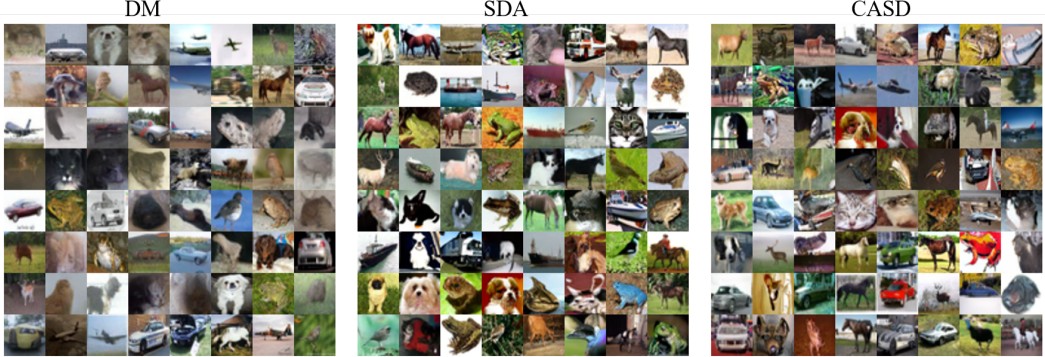

Figure 17: Generated samples from DM, SDA, and CASD trained with 1,000 real images and 10,000 synthetic images (Case 2 in Section 4.3)

DM                          SDA                          CASD

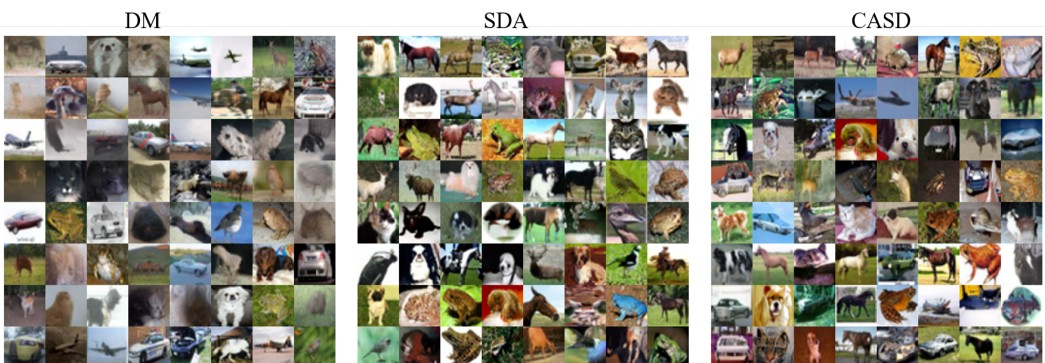

Figure 18: Generated samples from DM, SDA, and CASD trained with 1,000 real images and 10,000 synthetic images (Case 3 in Section 4.3)

DM                          SDA                          CASD

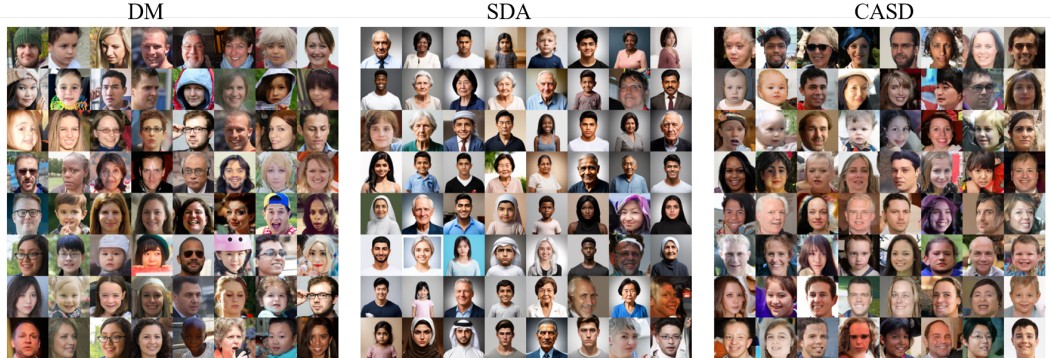

Figure 19: Generated samples from DM, SDA, and CASD trained on 1,000 real FFHQ images and 10,000 synthetic images generated by SD3.

DM                          SDA                          CASD

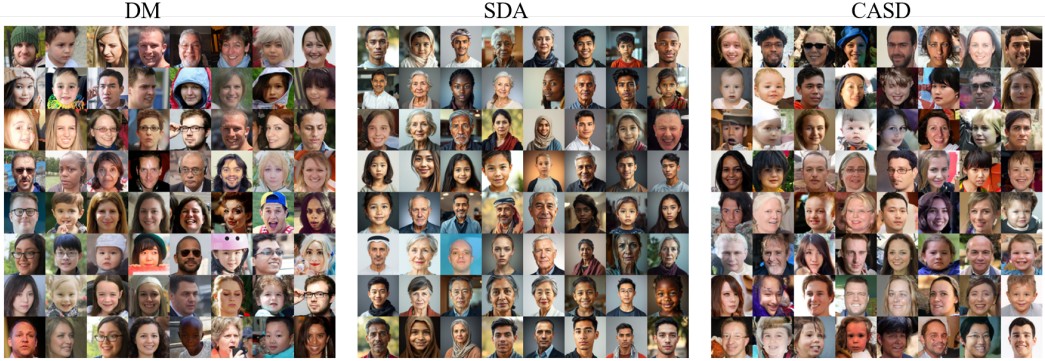

Figure 20: Generated samples from DM, SDA, and CASD trained on 1,000 real FFHQ images and 10,000 synthetic images generated by FLUX.

## C   THE USE OF LARGE LANGUAGE MODELS

Large language models (LLMs) were used in two limited ways in this work. First, LLMs were employed as a general-purpose writing assistant for minor text polishing and grammar correction. Second, LLMs were used to construct diverse text prompts for generating synthetic human-face images in our experiments. No part of the research design, methodology development, data analysis, or experimental implementation relied on LLMs.

