# OpenReview forum: "Conditional Augmentation Enables Effective Use of Synthetic Data in Diffusion Models"
_ICLR.cc/2026/Conference — ICLR 2026 Conference Withdrawn Submission_

### Official Review · Reviewer_xHB6 · 2025-10-23

**Soundness:** 3
**Presentation:** 3
**Contribution:** 3
**Rating:** 6
**Confidence:** 3

**Summary:**

The paper provides a simple, yet effective method for synthetic data augmentation in diffusion models: when training the model, give it a label as to whether the data is real or synthetic. This gives the benefit of increasing the training data for how to extract features (from synthetic images), while avoiding collapse that could come from overfitting to synthetic data (because the model is told that it is fake). When they go to generation, they simply just pass in the real conditioning label to the model to tell it to generate real images.

**Strengths:**

-The method is intuitive.
-The method works, and is easy to implement. The simplicity would suggest that it has potential for wide adoption.
-The experiments show improvement over baseline in a comprehensive variety of tasks, where other methods failed (e.g., increasing the amount of synthetic data by a lot).
-The paper is well-written and easy to follow.

**Weaknesses:**

- The paper only tests on CIFAR and FFHQ. These are relatively small and undiverse datasets. Results would be more convincing with experiments on ImageNet. (But if time does not allow, the authors can try CelebA, etc.)

**Questions:**

- The method reminds me of the leakage-free data augmentation in EDM [1]. It could be good to discuss the links to that.

[1] Karras, Tero, et al. "Elucidating the design space of diffusion-based generative models." Advances in neural information processing systems 35 (2022): 26565-26577.

---

### Official Review · Reviewer_6zUM · 2025-10-26

**Soundness:** 2
**Presentation:** 1
**Contribution:** 2
**Rating:** 2
**Confidence:** 5

**Summary:**

This work introduces Conditional Augmentation with Synthetic Data (CASD). The method involves labeling training examples as either "real" or "synthetic" and then training a score network that is conditional on this source label. During inference, the condition is fixed to the "real" domain ($y=0$), directing the model to generate samples that follow the target (real) data distribution. The authors claim this allows the model to leverage representations that have been strengthened by the synthetic data. The method is evaluated on low-resolution datasets (CIFAR-10 $32 \times 32$ and FFHQ $64 \times 64$) and benchmarked against two baselines: (i) DM, an unconditional model trained solely on real data, and (ii) SDA, an unconditional model trained on a naive mixture of real and synthetic data.

**Strengths:**

The CASD framework is straightforward and simple to implement . Its core method of using source labels to train a conditional model provides a direct approach to separate the target distribution from the augmentation data .

**Weaknesses:**

## Weakness in Problem Formulation

The paper aims to answer how synthetic data can be used effectively without degrading model quality. However, the motivation and experimental setup provided do not clearly support the paper's claims.

The paper cites "model collapse" as a primary motivation. This is misleading. The model collapse (or MAD) literature addresses a recursive problem where models are trained on unlabeled web data that is already "contaminated" with synthetic samples. The central difficulty in MAD is the infeasibility of perfectly filtering this data at scale. In contrast, this paper assumes an "oracle" setting, where a perfect "real" vs. "synthetic" label is available for every single sample. This premise sidesteps the core challenge of the model collapse problem.

The analogy to LLM self-improvement is also flawed. In the LLM domain, synthetic data (like Chain-of-Thought) is typically highly structured, curated, and selected via a verifier or reward signal to optimize a specific task (e.g., reasoning), a paradigm closer to reinforcement learning. This often intentionally shifts the model's output distribution. The paper, however, seems to have an implicit goal of improving fidelity to the real data distribution, which is not explicitly formalized and is a very different goal from the task-optimization paradigm it references.

The experimental setup, which uses a limited amount (e.g., 10k samples) of synthetic data from a "stronger" model alongside a subset of real data, is ambiguous and poorly justified.

- If the generative model for the synthetic data is available (as it is in the experiments), it is unclear why one would be limited to only 10k samples, as one could sample far more. This makes the "limited" premise feel artificial and is inconsistent with the paper's own methodology of generating new synthetic data.
- If this "limited" set is meant to represent contaminated data from the internet, the premise fails, as it returns to the first point: it is practically impossible to perfectly separate this data into a "synthetic subset" in the wild.
- If the goal is to transfer knowledge from the "stronger" model to a new model trained on limited real data, the problem is more accurately described as model distillation. However, the paper does not frame its contribution this way and fails to compare against any standard distillation methods.

A critical experiment is missing from Table 1. If the paper's claim is that this method can improve a model beyond what is achievable with the real data alone (i.e., true self-improvement), then the most important baseline is missing: training on the full real dataset (e.g., $n=50k$ for CIFAR-10). This is the only setting that could show if CASD can enhance a model that is already "saturated" with real data. Its omission is a significant flaw.

Moreover, in this "full-data" regime, where the claim of true self-improvement is tested, the paper should be comparing itself to other relevant works that also use a model's own synthetic data for improvement, which are not mentioned [1,2,3,4].

## Limited and Impractical Experimental Scope

The paper's validation is narrow, leaving its practical utility in question.

**Simple Datasets:** The evaluation is restricted to low-resolution datasets (CIFAR-10 $32 \times 32$ and FFHQ $64 \times 64$). This provides insufficient evidence that the method is applicable or effective for modern, high-resolution generative modeling.

**Unconditional Generation Only:** The paper only explores unconditional (or binary source-conditional) generation. This ignores the most common and practical use case: class-conditional generation. It is unclear how CASD would be applied in this setting. For example, on CIFAR-10, would the model require a separate "synthetic" label for each class (e.g., "real dog" vs. "synthetic dog")? Or would a single "synthetic" label be added, conflating all synthetic classes? The paper offers no guidance on this, making its practical applicability ambiguous.


[1] Yuan H, Chen Z, Ji K, Gu Q. Self-play fine-tuning of diffusion models for text-to-image generation. Advances in Neural Information Processing Systems. 2024 Dec 16;37:73366-98.
[2] Alemohammad S, Humayun AI, Agarwal S, Collomosse J, Baraniuk R. Self-improving diffusion models with synthetic data. arXiv preprint arXiv:2408.16333. 2024 Aug 29.
[3] Zheng K, Chen Y, Chen H, He G, Liu MY, Zhu J, Zhang Q. Direct discriminative optimization: Your likelihood-based visual generative model is secretly a gan discriminator. arXiv preprint arXiv:2503.01103. 2025 Mar 3.
[4] Kim D, Kim Y, Kwon SJ, Kang W, Moon IC. Refining generative process with discriminator guidance in score-based diffusion models. arXiv preprint arXiv:2211.17091. 2022 Nov 28.

**Questions:**

- Have authors tested the setting where n=50k, using full real dataset to see if the performance still increases?

---

### Official Review · Reviewer_4xzD · 2025-10-30

**Soundness:** 2
**Presentation:** 2
**Contribution:** 2
**Rating:** 2
**Confidence:** 4

**Summary:**

This paper aims to use synthetic data to improve diffusion model performance in limited data scenarios. The method combines synthetic and real data by assigning a binary label that indicate whether a data sample is synthetic or real. A diffusion model is trained on this combined dataset using the binary label as conditions. During inference, the binary label condition is set to real. Images generated by this approach have lower FID compared to directly training the diffusion model on real data and combination of real and synthetic data.

**Strengths:**

- The idea of this paper is easy to follow
- The approach can improve the quality of synthetic data.

**Weaknesses:**

- Limited experiments. The authors should use more benchmarks such as Caltech101, Oxford-IIIT Pet, Stanford Cars. Two small datasets with 32x32 and 64x64 resolution cannot sufficiently demonstrate the performance of the proposed method.

- Baselines are missing. The authors do not compare with other synthetic data generation pipelines using diffusion models, such as GIF-SD, DiffuseMix, DistDiff

- Limited significance. Only image generation benchmarks are evaluated. Although images generated by the proposed method has lower FID, there are no evidence that the synthetic images generated by the proposed method can truly improve desired task performance such as image classification. There should be evidence and discussion on how the synthetic data generated are impactful.

- There are some writing mistakes and typos. For example, in Line 182, the conditional distribution should be $\tilde{P}_{X|Y}$ otherwise it means training with real data only. The authors should check equations and math carefully.

References

Expanding Small-Scale Datasets with Guided Imagination (NeurIPS 2023)

Diffusemix: Label-preserving data augmentation with diffusion models (CVPR 2024)

Distribution-Aware Data Expansion with Diffusion Models (NeurIPS 2024)

**Questions:**

The proposed approach uses synthetic data generated from other models to generate additional synthetic data with lower FID. I am concerned on the total amount of resources required. For example on a new dataset without already generated synthetic data, the proposed method requires a generative model to first generate synthetic data, and then use this synthetic data to train a diffusion model to improve the quality of synthetic data. Have the authors considered how to directly generate synthetic images from real data?

---

### Official Review · Reviewer_xviM · 2025-11-06

**Soundness:** 2
**Presentation:** 4
**Contribution:** 3
**Rating:** 4
**Confidence:** 3

**Summary:**

This paper addresses the evergreen problem of how to integrate synthetic data into the learning pipeline

**Strengths:**

The key idea of the paper is, rather than integrating the real and synthetic data into one whole, the data gets labeled as either real or synthetic, and then the modeling process makes use of this information, as outlined in section 3. Overall, I think this part is very well explained, and a novel idea I have not seen before. The writing is high quality, and it does address a lacuna in the literature.

While I had some issues with the evaluation, as outlined below, figures 2 and 3 gave good insight into the method as compared to the baseline approaches.

**Weaknesses:**

The main weakness of the paper lies in the evaluation. Table 1 shows the main experimental results, which are further elaborated on in figures 2 and 3. While mostly promising, Cifar images are 32x32, and the FFHQ images are downsampled to 64x64. Both of these are tiny resolutions by the modern standards of generative image models, and remain not convincing that this will work as well at larger resolutions.

Further, the paper overstates its achievements in several places. For example, the authors write as one of their 3 contributions that “We articulate three fundamental principles that formalize the systematic use of synthetic data in generative learning”. However, these so called principles, and especially the first 2, are ideas that have been already articulated in hundreds of papers on synthetic data. Also, it is not clear how the third principle, namely broad applicability, is achieved in a paper focusing on diffusion models.

**Questions:**

How would this work be broadly applicable? Can you use this outside of the narrow domain of the paper? This was claimed to be the case, but I do not see how.

Can you demonstrate this will work for full-resolution images?

---

### Note · Authors · 2025-11-12

I have read and agree with the venue's withdrawal policy on behalf of myself and my co-authors.